# Juggling between caregiving and self-actualization: Older parents' lifelong experience of caring for an adult child with developmental disabilities

**Hila Avieli** [1], **Tova Band-Winterstein**[2], **Alon Zamir** [3]*

**1** Department of Criminology, Ariel University, Haifa, Israel, **2** Department of Gerontology, The Minerva Center on Intersectionality in Aging (MCIA), University of Haifa, Haifa, Israel, **3** Department of Community Mental Health, The Minerva Center on Intersectionality in Aging (MCIA), University of Haifa, Haifa, Israel

* alon6891@gmail.com

## Abstract

Recently, the number of adults with Developmental Disabilities (DD) who live with their parents has increased. This study aims to explore how parents report retrospectively and interpret their experience in the context of self-actualization in the long-term care of a child with a developmental disability. Four forms of parents' experiences emerged from the analysis: "This child is my whole world"–Total devotion; "I Can Do Both"—Actualizing personal and familial goals as well as caregiving issues; "It's a mission, it's a calling, it's a full-time job"–Self-actualization through caregiving; and "Disability will not stop me"—Emphasizing self-actualization. While prior studies have created a distinct separation between caregiving and self-actualization, the current study focuses on the complex dynamics of lifelong parental caregiving for a child with DD, illustrating the parents' ways of actualizing their life goals in the context of caregiving over the years.

## Introduction

The birth of a child with Developmental Disabilities (DD) is considered to be a major life trajectory, which challenges the parents' life world [1–3], and has a profound effect on their well-being [4], and on the utilization of social and professional opportunities [5, 6]. In this context, changes in parents' personal goals may be necessary in order to provide for their child's care [7–10]. The aim of the present study is to explore how parents retrospectively experience and interpret *self-actualization*, which is defined as "the desire to become increasingly what one is, to become everything that one is capable of becoming" [11, p. 382] in the context of long-term caregiving to a child with developmental disability (CWDD).

### Parental caregiving for a child with a developmental disability along the life course

Parental caregivers play a critical role in the health and long-term care system by providing a significant proportion of the necessary care to individuals with DD [12–14]. The impact of

(meraz@netvision.net.il) for researchers who meet the criteria for access to confidential data.

**Funding:** The study was supported by: The Shalem Fund for Development of Services for People with Intellectual Disabilities in the Local Councils. grant number: 123/2017

**Competing interests:** The authors have declared that no competing interests exist.

caregiving on parent-caregivers has received considerable research attention [15, 16]. Findings confirm that stress is a prevalent feature across the life course of the parental caregiver [17]. It is suggested that caregiving may lead to sleep disturbances [18–20], fatigue [21], loneliness [22], and social isolation [22, 23]. Parental caregiving may result in less emphasis on personal growth [11], centering family interactions and overall activities around the CWDD, and far-reaching career compromises. These descriptions of parents' stress and loss have led to the notion that parental coping with disability has been 'over-pathologized [24], and studies show a one-sided picture of what parents experience on a day-to-day basis. Hence, another body of research has emerged, revealing a more compelling look at parental caregiving. These studies show how caregivers can derive a sense of satisfaction, pride, and gratification from the care provider role [25–30]. Although parental caregiving has received considerable scientific attention, little has been written regarding how parents view their caregiving roles in the context of attaining other life goals and additional vocational activities along the life course.

## Beyond caregiving: Parents' aspirations and wishes in the context of raising a child with disability

In the parent-caregiver literature, achieving personal goals, which are not connected to the CWDD and their needs, is often perceived as unobtainable [31–33]. In this context, Todd and Shearn's [33] seminal research differentiated between 'captive' and 'captivated 'parents; that is, parents who experienced parenting a CWDD as restrictive and desire the lifestyle of their peers, and those who gave up their personal aspirations and ambitions and find positive meaning in their role as parents. Walden, Pistrang, and Joyce [34] further suggested that this may be a continuum, but similarly emphasized the contrast between these identities and the struggles between them. Evidence regarding parents' aspirations for themselves suggest that some parents of children with disabilities struggle to achieve a balance between parenting and caring roles; to maintain their personal identity through work, interests, and personal relationships; and to maintain family life, including their relationship with their spouse [35].

In this context, it is possible to use the concept of *self-actualization* to describe parents' aspiration for themselves, their vocation, and their own goals and wishes [36]. The classic concept of *self-actualization* is defined by Abraham Maslow [37] as: "the desire to become increasingly what one is, to become everything that one is capable of becoming" (p. 382). He suggested that self-actualization may be the highest level in the hierarchy of physiological and social needs [38]. Any attempt at defining what constitutes self-actualization, then, is but an articulation of one's subjective values and ideals of what is good, healthy, and desirable [39]. Moreover, contemporary perceptions of self-actualization have broadened the scope of this concept and claim that it may be derived from multiple and independent motivational systems, and that the individual prioritizes and shifts self-actualization needs throughout the life span [40, 41].

In applying the concept of self-actualization to parenthood, its complexity is further elucidated: On the one hand, for many people becoming a parent is perceived as a way of self-actualizing one's potential, emphasizing feelings of fulfillment, personal growth, and accomplishment [42]. On the other hand, parenthood might at times be a buffer between the parent and professional or personal goals; therefore, it is sometimes perceived as an obstacle hindering the achievement of self-actualization [40, 43].

Parents who provide care for children with disabilities report similar discrepancies regarding this issue. While some parents report feelings of self-actualization as a result of providing care for a CWDD [44], others often mention the disruptions of their life world due to rearing a CWDD, including giving up career opportunities [45], centering their lives around the CWDD, and the loss of hope for a better future [46]. Moreover, most studies have explored

specific subjects such as career involvement [47], or out of home placement, emphasizing a certain issue in a specific phase in the family's life, for example: career development of parents of young children with disabilities [48] or physical recreation of parents to children with disabilities [49]. However, to the best of our knowledge, most of the studies did not involve parents' retrospective view of self-actualization (e.g., parents' experience that they are, in fact, becoming everything that they are capable of becoming) in the context of their child's disability.

### Parents' retrospective view of lifelong caregiving

*Caregiving* has been described as identity-defining in the way it changes the caregiver's life [26, 50]. Some parents have described losing parts of their former identity due to the domination of the parent-caregiver role [35], and view it as an overwhelming experience, but also as a source of self-meaning [33]. Therefore, the ways in which parents view themselves and their roles, and how these roles intersect and conflict with each other, are central to understanding parents' perspective of their own lives. This may become even more true when parents and children with disabilities grow older [51]. Evidently, parents of adult children with disabilities are occupied with questions regarding the meaning of their lives as parents and caregivers, and the way they have shaped meaning through the natural process of life review [51, 52].

Contemplating self-actualization is a significant issue in the life review process, in which parents reflect and give retrospective meaning to past experiences and choices, in light of their current life's consequences [11, 53]. Years of intense caregiving shapes and colors parents' identity throughout the life course [54]. Life review, includes looking at the past and examining their choices [55]. Thus, parents of older adults with disabilities may reflect on choices that were made which affected their caregiving to the CWDD, as well their self-actualization and the interplay between these two concepts.

## Method

An "interpretive phenomenological analysis" (IPA) method was used, as this method is well-suited to studying the psychology of health and illness, and is widely used in disability studies [3, 42, 56].

Committed to the examination of how people make sense of major life experiences [3, p.1], IPA adopts an interpretive ontological stance. That is, it attempts to uncover the meaning and, in turn, the reality of people's experiences in the social world. IPA does not view reality as objective, nor is it a methodological approach at the positivist end of the ontological continuum. It does not attempt to define or indeed obtain facts or seek the truth. Rather, it seeks to understand the person's own experience, the meaning they make of it and, crucially/most importantly, the interpretation which the researcher makes of the person's meaning. Smith et al [3] describe IPA as 'an approach to qualitative, experiential and psychological research, which has been informed by concepts and debates from three key areas of philosophy of knowledge: phenomenology, hermeneutics, and ideography' (p.11). IPA draws on each of these theoretical approaches to inform its distinctive epistemological framework and research methodology [57].

The Journal Article Reporting Standards for Qualitative Research [58] which outlines key standards for qualitative researchers, was adhered to throughout the data collection and analysis.

### Procedure

The research team included three researchers; two certified social worker researchers with a PhD in gerontology and criminology; and a researcher in the field of special education with a

PhD in gerontology. The three researchers have extensive experience in qualitative research on sensitive topics. The research was approved by the University review board (IRB), as well as by the ethical board of the Welfare Ministry, which oversees the conduct of Non-Governmental Organizations (NGO) involved with individuals with disabilities. This approval enabled the research team to reach out to NGOs using two main strategies: one strategy included reaching out to NGO professionals, explaining the study's aim, and receiving a list of potential participants. Another strategy entailed presenting the research topic to groups of parents and their adult children with disabilities during NGO meetings. After contacting NGOs and introducing the research, a letter was distributed to parents inviting them to take part in the research. Subsequently, the researchers were contacted by those parents who had an interest in participating in the study. Potential participants consented to having their contact information given to the researchers, after an introductory telephone call from their case manager at the NGO. Only parents who gave their initial consent were approached by the researchers. Face-to-face interviews took place at the participants' location of choice—in most cases, in the participant's home. The interview began with the researcher introducing herself once again and explaining the research aims. The duration of the interview depended on the individual needs of the participants and usually lasted from one to two hours. Considering the preparation phases of the procedure—which included contact via NGO professionals, for each interview, including preliminary telephone calls and presentations to parents—the response to the interview was very positive and we managed to conduct 16 interviews with no dropouts.

In order to protect the anonymity of the participants pseudo names were used to describe the participants and their families. In addition, in some cases the gender of the participants child was changed as well to provide anonymity (for example: the child with disability is presented in the table as a male when in fact, she is a female).

## Participants and sample

Fifteen parents (8 fathers and 7 mothers) aged 61–89 (mean age: 72.8), participated in the study. The participants were purposefully selected [59] by criterion sampling, to obtain the widest possible variation of respondents among parents to an adult CWDD. In this context, we used a sampling chart in order to include various living arrangements, occupational statuses, and marital statuses. Inclusion criteria were as follows: participants had to be over 60 years of age, with no cognitive deterioration, and good verbal capabilities. All participants are parents of adult children with developmental disabilities. The age range of the adult children with disability was 25–62, half of whom reside at home. Adult children were diagnosed with three types of developmental disabilities: cerebral palsy (4 children), autism spectrum disorder (6 children), and intellectual disability (5 children). The final sample size (15 parents) was determined according to the principle of 'theoretical saturation' [60], which claims that the sample size in phenomenological studies is determined by the richness and depth of the data gathered from the informants. Morse links saturation with the idea of replication [61], and claims that "the domain has been fully sampled–when all data have been collected–then replication of data occurs and with this replication. . . the signal of saturation" [60: p. 148]. In this context, it has been suggested that the more information the sample holds, relevant to the actual study, the lower number of participants is needed [62]. Thus, in the current study, after 15 participants were interviewed, recurring content indicated that saturation had been reached.

See Table 1 for participants demographics

**Table 1. Participants.**

| | Family Unit | *Name | Age | Family status | Residence |
|---|---|---|---|---|---|
| 1 | Inbar | Father: Abraham | 64 | Married | Family home |
| | | **CWD: Ben | 32 | Single | Family home |
| 2 | Kaplan | Mother: Lea | 75 | Widow | Family home |
| | | CWD: Gil | 38 | Single | Independent |
| 3 | Brown | Father: Joshua | 66 | Married | Family home |
| | | CWD: Alex | 37 | Single | Family home |
| 4 | Lee | Father: Josef | 63 | Married | Family home |
| | | CWD: David | 25 | Single | Family home |
| 5 | Baker | Mother: Elizabeth | 89 | Widow | Family home |
| | | CWD: Julie | 58 | Single | Family home |
| 6 | Green | Mother: Libby | 64 | Married | Family home |
| | | CWD: Isaac | 38 | Single | Family home |
| 7 | Smith | Mother: Ruth | 63 | Married | Family home |
| | | CWD: Mike | 40 | Single | Family home |
| 8 | Gray | Father: Max | 66 | Married | Family home |
| | | CWD: Tom | 37 | Married | Independent |
| 9 | Simpson | Mother: Helen | 61 | Married | Family home |
| | | CWD: Rebecca | 25 | Single | Family home |
| 10 | Hunter | Father: Dan | 80 | Widower | Family home |
| | | CWD: Ron | 58 | Married | Independent |
| 11 | Zilbar | Mother: Liz | 87 | Widow | Family home |
| | | CWD: Ethan | 62 | Married | Independent |
| 12 | Stone | Mother: Rachel | 85 | Widow | Family home |
| | | CWD: Maya | 60 | Married | Independent |
| 13 | Scott | Mother: Sara | 65 | Divorced | Family home |
| | | CWD: Jacob | 42 | Single | Independent |
| 14 | Gonzales | Father: Solomon | 75 | Widower | Family home |
| | | CWD: Jonathan | 40 | Single | Family home |
| 15 | Weizmann | Father: Moshe | 70 | Married | Family home |
| | | CWD: Rose | 38 | Married | Independent |

*The names used in this table are pseudo names. In addition, we changed the gender of some of the CWDs.

**CWD: child with disability

## Data collection

The interview guide included four main content categories: family relationship throughout the years with a CWDD; day-to-day life with disability over the years; caregiving routines and roles; dream and wishes, (vocational and others)–what was achieved and what was not; and reflecting on the lifelong experience of parenthood to a CWDD.

## Data analysis and trustworthiness

Data analysis was performed according to the interpretive phenomenological analysis (IPA) method [3]. All researchers were involved in analyzing the data. First, each researcher read the transcripts several times in order to become as familiar as possible with the text. At this stage, each reading includes pinpointing significant statements, for example: identifying a wide range of parental feelings about adult children with DD. The next step involved grouping the

statements into themes, including quotes that capture the essential quality of the participants' experiences and perceptions, for example, gathering quotes relating to the parents' role in the child's life, the parents' other aspirations, and their feelings regarding them. The following step involves identifying the emerging connections, clustering them together, and conceptualizing them. During the textual analysis, the researchers discussed and identified the ways in which participants' accounts were similar, but also different, i.e., most parents viewed themselves as being dedicated to providing care for their children with disability, but each parent reframed and constructed caregiving differently. In this stage, we heuristically used the concept of self-actualization in a deductive manner in order to provide a broader understanding of the findings and enable the identification of different parental experiences.

The data was organized based on four agreed-upon patterns of parents' experiences [3]. *Trustworthiness* was achieved through the use of reflexivity [63]. We established "bracketing," by reflecting on our experiences, biases, and prejudices regarding disability throughout all of the study's stages [64]. Member checking was used throughout the process individually with each participant. Adherence to this procedure enhanced the study's credibility [65].

## Ethical considerations

Special attention was paid to the emotions expressed by the participants. As interviewers, we utilized our professional experience to create an atmosphere in which the interviewees felt comfortable sharing candid information [42]. Special provisions were made to ensure informed consent and confidentiality [42]. Following the university's IRB approval, each participant signed a written informed consent form. Freedom to refuse to continue or withdraw from the study at any point was emphasized. In addition, the interviewer continuously sought process consent for those informants who appeared distressed during the interview [66]. To ensure anonymity, any potentially identifying information was removed from this paper. In addition, interviews were saved on the author's personal computer, which allowed no outside access.

## Findings

Living with disability over the course of a lifetime reveals an inner dialogue between the desire and obligation to dedicate oneself to the CWDD, and the need to fulfill personal aspirations, such as hobbies, professional goals, vocational ambitions, familial goals, and other hopes and dreams that are not related to the CWDD. Four forms of parental experiences emerged from the analysis: "This child is my whole world"–Total devotion; "I Can Do Both"—Actualizing personal and familial goals as well as caregiving issues; "It's a mission, it's a calling, it's a full-time job"–Self-actualization through caregiving; and "Disability will not stop me"—Emphasizing self-actualization.

### "This child is my whole world"–Total devotion

Some of the parents in the study devoted their lives to caring for their child, interrupting almost all other activities in the process, as illustrated in the following quotes:

*It's a big question—whether to be one hundred percent for the child, sacrifice everything, or leave something for yourself as well. I am a second-generation Holocaust survivor; the issue of parenthood and doing everything for the child is in my blood. So, I put everything else aside, because I knew that I couldn't survive, unless I gave it my all. . . One day, I decided to commit suicide. At that moment, I felt that I didn't really exist anymore. I was Rebecca's mother, a wife, and a caretaker; but me, myself—I was just gone, vanished. . . It took years of therapy to*

*get over it. . . I realize now, at my age, that I can't go on living this way. This child is my whole world. Caring for her is all I did for years, and somehow, I lost myself. . .*

(Helen, 61)

Helen demonstrates the ultimate self-dedication, almost to the point of complete self-renunciation. Her explanation for this approach is rooted in her family history as a Holocaust survivor. This personal history is reflected in Helen's choice to diminish herself as a person with ordinary needs, for the benefit of her CWDD. The extreme expression of this parental approach is expressed by the desire to end a life that has lost all sense of meaning. It seems that at this point of her life, the retrospective view leads her to engage in soul-searching and to question the meaning of her choices over the years.

The next quote illustrates another version of total devotion:

*I would come home from work and go straight to him (to the CWDD), where else would I go? I take care of him, wash his face, take him to the bathroom, give him a shower, everything. And the rest of the kids, they have to manage on their own. They felt I gave all my love to Ron and loved them less. . . my wife—she sleeps alone, that's the deal. . .Nowadays, I sleep with Ron at night, he needs me now even more than when he was younger.*

(Dan, 80)

This parent devoted all his life to the total care of his CWDD, even though he was aware of the pain it caused his wife and other children. Now, in old age, when the child is already over 50 years old, and his needs have changed due to the aging process, the father still emphasizes his child's needs, ignoring his own aging process.

## "I Can Do Both"–Actualizing personal goals as well as caring for the child with developmental disability

Another form of parental experience related to providing lifelong care to a CWDD is expressed by dedicating oneself to the child and also perceiving the actualization of one's personal goals as life missions:

*I guess I am a total giver. . .after the twins were born, I decided to open my own beauty parlor; I wanted to have something for me, something that would be a success. Everyone thought I was trying to run away from home; they said my children would be neglected, but I knew I could do both, no compromises, no regrets. My babies had a devoted mother every day from noon until night. . .and then everyone asked "How do you do it all?". I understood over time that the more you do, in all aspects of life, the more energy you have. . .. Today, I can see the benefits, but I'm very tired. . .*

(Libby, 64)

The reality of raising and caring for baby twins, one of whom was diagnosed with severe DD, challenged this mother's familial and personal goals. As a "giver", she deliberately chose to invest both in her children and her career, without allowing one to take precedence over the other. She consistently proved to herself and to others that she could successfully be both a devoted mother and businesswoman. It seems that this way of life became a key theme and a source of motivation and energy in her general life perception. This explains and accounts for past choices. Retrospectively, she starts to acknowledge both the empowering advantages of

this approach, and also the costs of being totally dedicated to both of these goals. This is expressed in her final statement regarding feeling tired as a result of doing both.

Another version of "I could do both" is illustrated by the following quote:

> *That's the thing you do all the time—you take care of your child, you take care of your family, and you make an effort to look your best all the time. I never give up, never cut myself any slack, but emotionally it runs you down. People who don't know me always tell me that I look happy, that I don't look like I have a child with disability, but if they take a closer look they see the truth. . . one day a female customer came up to me and said: 'You know, you are lovely, and you radiate an angel's glow. . .it's probably god's gift, but your eyes are sad. If you look past the beautiful outer shell, you can see it. You are like a sad clown'. That's what she said and she was right. I'm very dominant and take care of everything on the outside, but I'm crying on the inside. . .*

(Elizabeth, 89)

The "sad clown" metaphor encapsulates the pain and the struggle of simultaneously maintaining both self-actualization and caregiving. In this stage of life, this mother reflects upon this approach and realizes the emotional price of being invested in her child, her family, and in her career for all these years.

## "It's a Mission, It's a Calling, It's a Full-time Job"–Self-actualization through caregiving

Parents in this category found ways to organize their lives around the disability, and create a sense of self-actualization:

> *I got countless job offers, including a full-time nanny for my son, and whatever I wanted, and then I decided to put all my energy into my son, and I never felt like a victim. It was an informed decision I made from the heart, and I've never regretted it. It does determine a lot, but you have to decide whether or not to immerse yourself in it. And I decided to take care of him: it's a mission, it's a calling, it's a full-time job and I'm not sorry. Now that I'm getting older, I can still say that it's given me so much more than I would have gotten from any other job. . .*

(Lea, 75)

Lea starts by emphasizing her ability to choose between taking care of her son and other job opportunities–it's important to her to note that this was a rational decision, made with serious consideration and knowingly preferred over other possibilities. It seems that self-actualization is an important value for Lea; thus, she perceived caring for her son as equivalent to pursuing any other career. By doing so, Lea gives meaning to the years spent caring for her child and is at peace with the decision.

Another version of this approach is expressed by parents who used the experience of dealing with disability as a sort of leverage to create their own family business. By doing so, they merged their dedication to the CWDD with other daily activities, such as developing a business which could serve as both a source of income as well as a path to self-actualization:

> *I had just opened my own architect's office, when Mike was born. One day I walked in, looked around, and thought: 'What the hell am I doing here? This is not for me; I have a more important mission in life now' and I went back to Mike. . . Real survivors are people who solve*

*problems. We, (my wife and I) don't sit around and wait for things to happen to us... We started our placement agency directly according to Mike's needs, and as he grew up, we developed more services for the disabled community. Now that he is grown, he wants to move out, so we went into housing solutions... now we're retired and feel satisfied that we have something to pass on to our children, a business that has real value, but is also based on what we believe in.*

(David, 73)

David demonstrates an active and flexible perspective to the question of caregiving and self-actualization. The presence of a CWDD challenged the parents' plans and demanded changes. In this way, David gave up his professional path and re-invented himself in a new career that combines the needs of his CWDD. This decision has become a significant mission for him and a source of livelihood. At this stage of life, the parents pass on a sort of legacy that is embedded in the family business.

## "This is Not the End of My Life, It Will Not Stop Me"–Emphasizing self-actualization

The parents in this category chose to actualize personal goals as a first priority, sometimes perceiving it as a major life task; caretaking for the CWDD was perceived as a secondary mission:

*From the moment he was born, I knew that this was not the end of my life, it would not stop me. After a few months, I pulled myself together and went back to work. It was important to me to work full time; it helped me clear my head from what was going on at home. Back then, I was torn. I felt guilty and even ashamed, but I couldn't stay at home with him... when I tried staying home with him, I felt I was withering, it wasn't right for him or for me... He got used to spending most of his days without me and when he moved out at the age of sixteen, the transition was smooth. Over the years, I've realized that going back to work made me a better mother. I was able to spend quality time with him when I was around, and I let real professionals take care of him.... Lately, he's been calmer and it's easier to be with him. Since I've retired, I spend much more time with him.*

(Sara, 65)

Life's circumstances threatened to change Sara's planned life script. She shares her inner world and the complex emotional dilemmas that stood in her way regarding the issue of dedicating her time to her son or to herself. She felt that positioning herself as a caretaker was not a suitable option for her, and the expression "I was torn" describes the torment associated with this issue. Thus, she felt compelled to focus on her own self-actualization. As the years went by, it seemed that what had once been an emotional compulsion became a well-thought-out ideology. This ideology is manifested by giving retrospective accounts that affirm this behavior, for example: the smooth transition of the child from home to a hostel, and letting real professionals take care of him. In retrospect, the good relations she shares with her son today is the outcome of having chosen the best path in the past. It seems that old age has provided a peaceful sense of closure through the compensation processes (e.g., spending more time with her son) that moderated her approach throughout the years.

The next quote further strengthens this approach:

*We focused on giving our child the best options, his siblings the best possibilities, and ourselves an option to go on living, and not allow the disability to determine our lives. The decision to*

*move Tom into a care facility at a young age was extremely hard. Yes, he is my son; and yes, he deserves to live; and yes, so do we. We deserve to live, too. People may think it's the easier choice, but it's not. Parting from your flesh and blood and having him move out the house is the hardest thing for a parent. . . Now we are starting our golden years and we want to enjoy retirement. We want to travel. I saw families traveling with an autistic child, but that's not for us. . . I look back on many things and [sometimes] say we didn't do enough, but all and all, I don't regret any major decisions, and I don't regret creating a life for ourselves.*

(Max, 75)

This participant narrates a split between dealing with the disability and other life domains. Today, he describes feeling at peace with his decisions for himself and his family; however the difficult decision to have the CWDD move away is described as both a very hash and unpopular decision–which didn't necessarily become easier to live with over the years.

In sum, it seems that parents' narratives regarding living together with a child with disability may be perceived along a continuum: while some parents perceive themselves dichotomously as being very dedicated to caregiving ("This Child is My Whole World"–Total devotion), or conversely, as self-actualizers ("This is Not the End of My Life, It Will Not Stop Me"–Emphasizing Self-actualization), others present other options along this continuum. The themes: "I Can Do Both"–Actualizing Personal Goals as Well as Caring for the Child with Developmental Disability, and "It's a Mission, It's a Calling, It's a Full-time Job"–Self-actualization through caregiving, represent ways of integrating living alongside disability and realizing personal and professional goals and aspirations.

See Table 2 for Theme summary.

## Discussion

The critical approach to disability studies emphasizes the importance of self-actualization in the lives of people with disabilities and their families. This approach argues that limitation should not be an obstacle to self-actualization. In the past, the attitude of the disabled person and family was based on the "Tragedy model", in which living with a disability was perceived as a life of suffering and dissatisfaction [55]. This study is in line with various studies from recent years which have focused on the positive aspects, together with the difficulties, of living alongside a person with disability. The analysis of the findings reveal four patterns of parental experiences regarding the issue of self-actualization alongside living with a CWDD: "This child is my whole world"–Total devotion; "I Can Do Both"—Actualizing personal and familial goals as well as caregiving issues; "It's a mission, it's a calling, it's a full-time job"–Self-actualization through caregiving; and "Disability will not stop me"—Emphasizing self-actualization.

**Table 2. Theme summary.**

| Theme Num | Theme name | Theme's major premise | Number of parents identifying with the theme |
|---|---|---|---|
| 1 | "This child is my whole world"–Total devotion | Parents who devote their whole life to caregiving, while forsaking all other aspects of life. | 4 |
| 2 | "I Can Do Both"—Actualizing personal and familial goals as well as caregiving issues | Parents who struggle to simultaneously maintain both self-actualization and caregiving. | 5 |
| 3 | "It's a mission, it's a calling, it's a full-time job"–Self-actualization through caregiving | Parents who realize personal and professional goals through acts of caregiving. | 3 |
| 4 | "Disability will not stop me"—Emphasizing self-actualization | Parents who prioritize the actualization of personal and professional goals over caregiving. | 3 |

The current study's findings provide an "insider's" perspective of the lived experience of parents and the different ways in which they view their path to self-actualization in the context of raising a CWDD over the years. Some parents perceive caregiving and actualization as separate entities, while others perceive these experiences as being intertwined. While prior studies have created a distinct separation between caregiving and self-actualization and have tended to view these concepts as dichotomous [67], the current study shows a more detailed view of the phenomenon.

One common type of parental view, which was also found in this study, relates to self-dedication to the CWDD–**"This child is my whole world"–Total devotion**. This type of caregiving has been portrayed as a central experience in the lives of some parents of children with DD [68]. Devoted sacrificial behaviors require individuals to forgo immediate and future pleasures and rewards in favor of certain pain, injury, and even death. It has been suggested that some social bonds enable individuals to suppress their own self-preservation goals and preferences when necessary, in order to prioritize and promote the well-being of another person [69]. In the case of raising a CWDD in a home-based setting, intense caregiving to the child was found to be a necessary condition for maintaining the child's well-being [53]. In the current study, the "total devotion" perspective creates psychological distress, which is defined as "the end result of factors that prevent a person from self-actualization" [70]. In this sense, it seems that self-actualization and caregiving may be almost contradictory terms, as care relates to psychological distress that may prevent an individual from achieving self-actualization. An opposite parental experience which, emphasizes self-actualization, is presented in our findings as the **"Disability will not stop me—Emphasizing self-actualization"** category. This dichotomy expresses a broad familial perspective on the DD phenomenon, trying to take into account the needs of other family members, as well as the parents' needs, and preferring the best professional care for the CWDD, in order to normalize the experience of being a family under these special circumstances. This echoes the notion of *community rehabilitation*, which emphasizes the role of professionals within the community setting that support the person with disabilities and their family, thus allowing the family to conduct a normative and productive life [45, 52, 56].

Using a parental perspective reveals a more complex picture in which parents managed to integrate self-actualization into the care experience. Even though some studies acknowledge caregiving as a career [71], they also point out that this career is neither appealing nor desired and does not always produce feelings of self-actualization [71, 72]. In our findings, the pursuit of a career was interpreted differently, and the disability was perceived as a platform or window of opportunity for achieving self-actualization as expressed by the parental approach: **"It's a mission, it's a calling, it's a full-time job"–Self-actualization through caregiving**. This approach is adopted by parents who identify with the mission of caretaking and consider it a way of achieving self-actualization. These findings are in line with the contemporary view of self-actualization, which postulates that people reconstruct their dreams, aspirations, and personal goals in accordance with life changes and trajectories [36]. Thus, parents in this category found meaning in the caregiving mission and integrated it into their perception of self-actualization [73]. In this context, it is worth quoting the well-known psychologist Erich Fromm [74], who claims in his book "The Art of Loving" that the ability to love another person constitutes self-realization and that love is created through care and devotion.

Parents, who over the years presented a polarized approach of self-actualization over caregiving, in this stage of their life suggest a more moderate attitude towards spending time with the CWDD, as presented in the: **"This is not the end of my life, it will not stop me"** category. These parents put intensive efforts into maintaining both aspects of life, and are unwilling to place one above the other. It seems that over the years parents represented by this category

reveal signs of burden of care [75] expressed by fatigue [76], sadness [75], and perceived decreased quality of life [10].

Reviewing their life brings parents to do some soul searching, reflect on their own needs, their selves, and their choices over the years, considering their parental obligation of lifelong caregiving. It is interesting to note that parents who dedicated their lives to their child now allow themselves time to travel and enjoy life, while those who chose to dedicate themselves to their careers attempted to balance the tension between caregiving and self-actualization by adopting certain changes in their lifestyle. It seems that by so doing, they enable self-acceptance and closure processes that are significant for the self [13]. Parents who, over the years, managed to integrate caregiving and self-actualization expressed satisfaction with their choice at this point in their lives as well.

## Limitations and recommendations for further study

This study uses a retrospective view of the participants' life course. Further research may benefit from a longitudinal approach that provides real-time data. In addition, the research did not differentiate between different types of DD severity and range of disability and function impairment. Future studies should consider focusing on specific disabilities and the ramifications for the parents. Our sample included mostly parents who are functioning and independent. Further research should focus on parents who are more vulnerable and frailer.

## Conclusions and practical implications

The research findings indicate there is no unified profile of parental perceptions of self-actualization while living and aging with a CWDD. The different narratives presented in this study can serve as a framework for interventions that will meet parents' needs at this stage of their life. Professionals (such as social workers, and psychotherapists) should pay special attention to parents who display signs of burden of care, and encourage them to try to develop and fulfill self-actualization needs, in order to facilitate closure processes. Life review processes [77] may be bridled to strengthen the existing understanding of the choices that were made.

In addition, we recommend creating a support group for aging parents of a person with CWDD. This type of group can provide parents with the support of other parents who are in the same situation. We believe these groups could serve as a platform for the critical issue of self- actualization alongside limitation.

In conclusion, this article emphasizes the importance of a sense of self-actualization among parents providing care to a person with DD. Despite the great importance of this topic, there is very little research on it. We call for greater emphasis to be placed on designing intervention programs that take this issue into account.

## Author Contributions

**Writing – original draft:** Hila Avieli, Tova Band-Winterstein.

**Writing – review & editing:** Hila Avieli, Tova Band-Winterstein, Alon Zamir.

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
