## [Decision Letter · Decision Letter 0]

23 May 2022

PONE-D-22-09051Juggling Between Caregiving and Self-actualization: Parents’ Lifelong Experience of Caring for a Child with Developmental DisabilitiesPLOS ONE

Dear Dr. Zamir,

Thank you for submitting your manuscript to PLOS ONE. Based on my own reading and comments from two experts (see below) I would consider your paper for publication following some minor revisions. You should have no problems addressing the comments from Reviewer #2. The paper already is in a good shape but could be improved following minor revisions.

We look forward to receiving your revised manuscript.

Kind regards,

Robert Didden

Academic Editor

PLOS ONE

Journal Requirements:

6. We note you have included a table to which you do not refer in the text of your manuscript. Please ensure that you refer to Table 1 in your text; if accepted, production will need this reference to link the reader to the Table.

Reviewers' comments:

Reviewer's Responses to Questions

**Comments to the Author**

1. Is the manuscript technically sound, and do the data support the conclusions?

Reviewer #1: Yes

Reviewer #2: Yes

2. Has the statistical analysis been performed appropriately and rigorously? 

Reviewer #1: Yes

Reviewer #2: Yes

3. Have the authors made all data underlying the findings in their manuscript fully available?

Reviewer #1: No

Reviewer #2: No

4. Is the manuscript presented in an intelligible fashion and written in standard English?

Reviewer #1: Yes

Reviewer #2: Yes

5. Review Comments to the Author

Reviewer #1: Thank you for doing this research. It is an interesting span of experts that is being used for this. The use of the word "form" throughout the paper can mean different things in English so it took a few reads to understand the meaning. I would have liked more upfront definitions of self actualization over the self promotion of the researchers in how they have done other studies.

Reviewer #2: I enjoyed reading this clear, concise, rigorous and transparent manuscript that addresses a gap in the literature.

Please attend to the following minor revisions that will enhance this strong manuscript.

Abstract

“This study aims to explore how parents retrospectively experience and interpret” should say “retrospectively report on and interpret their experience.”

Title

Clarify that these are older adult parents of adult children.

Literature

Thorough. Self-actualization is well-defined.

Methods

“Method” is identified as IPA. Please describe and define IPA in two or three sentences.

Missing is overall research approach. What are the epistemological and ontological assumptions (paradigm), theoretical perspectives, and methodology (research tradition)? The methods flow from this along with the research question. See attached summary.

Ethics

“receiving a list of potential participants” from NGO. Did potential participants (service users?) consent to having their name/contact information given to researchers?

Table 1: Did the participants confirm that including those details would not identify them or they were agreeable to those details included? Is the population large enough to disguise individuals with these characteristics?

Analysis

Good description. Transparent. Inductive and deductive acknowledged.

What does member checking mean and how was it operationalized?

Findings

Please include summary at the beginning or end of this section with one to two sentence describing and comparing the four themes.

Include a figure of the four themes and their relationships/comparisons/contrasts.

Please include one quote from each of 15 participants.

Please clarify and put in Table 1 which parents fit which category and if they fit more than one. Was there any other pattern noted?

Further clarify differences between total devotion and self-actualizaton through caregiving.

Discussion

Further dialogue with the literature is needed.

Expand on the implications and relate to current interventions. Which professions would they impact?

Include a conclusion; article ends abruptly.

Writing

Minor spelling, grammar, punctuation for final review after revisions.

References

Consider removing Google Books links.

Some authors are missing.

Consider additional references on positive parental experiences:

Beighton, C., & Wills, J. (2017). Are parents identifying positive aspects to parenting their child with an intellectual disability or are they just coping? A qualitative exploration. Journal of Intellectual Disabilities, 21(4), 325–345. https://doi.org/10.1177/1744629516656073

Green, S. E. (2007). “We’re tired, not sad”: Benefits and burdens of mothering a child with a disability. Social Science and

Medicine, 64(1), 150–163. https://doi.org/10.1016/j.socscimed.2006.08.02

Matthews, E. J., Puplampu, V., & Gelech, J. (2021). Tactics and strategies of family adaptation among parents caring for children and youth with developmental disabilities. Global Qualitative Nursing Research: https://journals.sagepub.com/doi/pdf/10.1177/23333936211028184

Schall, C. (2000). Family perspectives on raising a child with Autism. Journal of Child and Family Studies, 9(4), 409–423. https://doi.org/10.1023/A%3A1009456825063

6. PLOS authors have the option to publish the peer review history of their article (what does this mean?). If published, this will include your full peer review and any attached files.

Reviewer #1: No

Reviewer #2: No

---

## [Author Response · Author response to Decision Letter 0]

15 Aug 2022

Thank you very much for your time and thought, attached is a letter with responses and corrections to all comments

---

## [Decision Letter · Decision Letter 1]

23 Aug 2022

PONE-D-22-09051R1Juggling Between Caregiving and Self-actualization: Older Parents’ Lifelong Experience of Caring for an Adult Child with Developmental DisabilitiesPLOS ONE

Dear Dr. Zamir,

Thank you for submitting your manuscript to PLOS ONE. After careful consideration, we feel that it has merit but does not fully meet PLOS ONE’s publication criteria as it currently stands. Therefore, we invite you to submit a revised version of the manuscript that addresses the points raised during the review process.

We look forward to receiving your revised manuscript.

Kind regards,

Robert Didden

Academic Editor

PLOS ONE

Journal Requirements:

Reviewers' comments:

Reviewer's Responses to Questions

**Comments to the Author**

1. If the authors have adequately addressed your comments raised in a previous round of review and you feel that this manuscript is now acceptable for publication, you may indicate that here to bypass the “Comments to the Author” section, enter your conflict of interest statement in the “Confidential to Editor” section, and submit your "Accept" recommendation.

Reviewer #1: (No Response)

Reviewer #2: All comments have been addressed

2. Is the manuscript technically sound, and do the data support the conclusions?

Reviewer #1: Yes

Reviewer #2: Yes

3. Has the statistical analysis been performed appropriately and rigorously? 

Reviewer #1: Yes

Reviewer #2: N/A

4. Have the authors made all data underlying the findings in their manuscript fully available?

Reviewer #1: Yes

Reviewer #2: No

5. Is the manuscript presented in an intelligible fashion and written in standard English?

Reviewer #1: Yes

Reviewer #2: Yes

6. Review Comments to the Author

Reviewer #1: Ethics Statement: Please write out what NGO stands for prior to only using its acronym. Readers outside of Israel may not know what this is.

Page 4: “his needs” should be changed to gender neutral phrasing.

Page 5: “his professional or personal goals” should be changed to gender neutral phrasing. The term his is used throughout and should be adjusted.

Page 6: run on sentence: Years of intense caregiving shapes and colors parents’ identity throughout the life course [51]; self-actualization may serve as the goal of life review, by creating closure and achieving self-acceptance for these parents [29].

Page 7: Please write out what NGO stands for at the first introduction of the acronym.

Table 1: were participants personal identifying information disclosed? I see this somewhat addressed in ethical considerations, however did you assign random names to participants?. How has the data collection and discussion remained anonymous to protect this population?

Page 18: Formatting on Discussion.

Page 24: Formatting.

Reviewer #2: Thank you for revising the manuscript according to all reviewers recommendations.

This article contributes to the literature in this field.

7. PLOS authors have the option to publish the peer review history of their article (what does this mean?). If published, this will include your full peer review and any attached files.

Reviewer #1: No

Reviewer #2: No

---

## [Decision Letter · Decision Letter 2]

14 Oct 2022

Juggling Between Caregiving and Self-actualization: Older Parents’ Lifelong Experience of Caring for an Adult Child with Developmental Disabilities

PONE-D-22-09051R2

Dear Dr. Zamir,

We’re pleased to inform you that your manuscript has been judged scientifically suitable for publication and will be formally accepted for publication once it meets all outstanding technical requirements.

Kind regards,

Robert Didden

Academic Editor

PLOS ONE

Additional Editor Comments (optional):

Reviewers' comments:

Reviewer's Responses to Questions

**Comments to the Author**

1. If the authors have adequately addressed your comments raised in a previous round of review and you feel that this manuscript is now acceptable for publication, you may indicate that here to bypass the “Comments to the Author” section, enter your conflict of interest statement in the “Confidential to Editor” section, and submit your "Accept" recommendation.

Reviewer #1: All comments have been addressed

2. Is the manuscript technically sound, and do the data support the conclusions?

Reviewer #1: Yes

3. Has the statistical analysis been performed appropriately and rigorously? 

Reviewer #1: Yes

4. Have the authors made all data underlying the findings in their manuscript fully available?

Reviewer #1: Yes

5. Is the manuscript presented in an intelligible fashion and written in standard English?

Reviewer #1: Yes

6. Review Comments to the Author

Reviewer #1: Thank you for resubmitting with revisions made. Additional research in this area will benefit many. I look forward to additional work.

7. PLOS authors have the option to publish the peer review history of their article (what does this mean?). If published, this will include your full peer review and any attached files.

Reviewer #1: No

---

## [Editor Report · Acceptance letter]

26 Oct 2022

PONE-D-22-09051R2 

Juggling between Caregiving and Self-actualization: Older Parents’ Lifelong Experience of Caring for an Adult Child with Developmental Disabilities 

Dear Dr. Zamir:

I'm pleased to inform you that your manuscript has been deemed suitable for publication in PLOS ONE. Congratulations! Your manuscript is now with our production department. 

Kind regards, 

on behalf of

Professor Robert Didden 

Academic Editor

PLOS ONE